# Biodegradation of Microtoxic Phenylpropanoids (Phenylpropanoic Acid and Ibuprofen) by Bacteria and the Relevance for Their Removal from Wastewater Treatment Plants

**DOI:** 10.3390/genes14020442

**Published:** 2023-02-09

**Authors:** Regina-Michaela Wittich, Ali Haïdour, Inés Aguilar-Romero, Jesús de la Torre-Zúñiga, Pieter van Dillewijn

**Affiliations:** 1Department of Environmental Protection, Estación Experimental del Zaidín CSIC, Calle Profesor Albareda 1, 18008 Granada, Spain; 2Unidad de Resonancia Magnética Nuclear, Centro de Instrumentación Científica, Universidad de Granada, Paseo Juan Osorio S/N, 18071 Granada, Spain

**Keywords:** emerging contaminants, ibuprofen mineralization, 3-phenylpropanoic acid mineralization, horizontal gene transfer, PPCP, NSAID

## Abstract

The NSAID ibuprofen (2-(4-isobutylphenyl)propanoic acid) and the structurally related 3-phenylpropanoic acid (3PPA), are widely used pharmaceutical and personal care products (PPCPs) which enter municipal waste streams but whose relatively low rates of elimination by wastewater treatment plants (WWTPs) are leading to the contamination of aquatic resources. Here, we report the isolation of three bacterial strains from a municipal WWTP, which as a consortium are capable of mineralizing ibuprofen. These were identified as the *Pseudomonas citronellolis* species, termed RW422, RW423 and RW424, in which the first two of these isolates were shown to contain the catabolic *ipf* operon responsible for the first steps of ibuprofen mineralization. These *ipf* genes which are associated with plasmids could, experimentally, only be transferred between other *Sphingomonadaceae* species, such as from the ibuprofen degrading *Sphingopyxis granuli* RW412 to the dioxins degrading *Rhizorhabdus wittichii* RW1, generating RW421, whilst a transfer from the *P. citronellolis* isolates to *R. wittichii* RW1 was not observed. RW412 and its derivative, RW421, as well as the two-species consortium RW422/RW424, can also mineralize 3PPA. We show that IpfF can convert 3PPA to 3PPA-CoA; however, the growth of RW412 with 3PPA produces a major intermediate that was identified by NMR to be cinnamic acid. This and the identification of other minor products from 3PPA allows us to propose the major pathway used by RW412 to mineralize 3PPA. Altogether, the findings in this study highlight the importance of *ipf* genes, horizontal gene transfer, and alternative catabolic pathways in the bacterial populations of WWTPs to eliminate ibuprofen and 3PPA.

## 1. Introduction

The efficient removal of pharmaceutic contaminants from effluents of wastewater treatment plants (WWTPs) and other water resources is eminently important, as ibuprofen in particular was demonstrated to be of high risk with respect to the availability of safe drinking water supplies [1]. Its widespread occurrence and distribution in the environment as well as advances in its (bio)remediation have been reviewed [2]. Recently, we showed the improvement of effluent decontamination of ibuprofen in a model wastewater treatment system by the ibuprofen mineralizing strain *Sphingopyxis granuli* RW412, thereby contributing more results from the genetic, as well as physiological and chemical points of view, of the catabolic pathway reactions [3] to advance knowledge on the biodegradation of ibuprofen.

Bacterial species capable of mineralizing ibuprofen have previously been isolated, and several aspects of their catabolism and genetic background have been elucidated. All hitherto isolated bacteria are found within the family *Sphingomonadaceae* (of the order *Sphingomonadales*) of the *Alphaproteobacteria*. The first ibuprofen-mineralizing, and not only co-metabolizing bacterial species, *Sphingomonas* sp. Ibu-2, was isolated in the USA, and several of the intermediary catabolic reactions for ibuprofen degradation were proposed [4]. The same authors showed that the *ipf* genes are responsible for the catabolism of this compound [5]. Almost identical catabolic genes within the above mentioned *ipf* operon were found to be associated with transposable elements and located on one of two megaplasmids of another ibuprofen mineralizing isolate, *Rhizorhabdus* (former *Sphingomonas*) *wittichii* strain MPO218, from which they could be transferred horizontally to another *Sphingomonadaceae* strain, *S. granuli* TFA [6]. Recently, several genes were proposed for the lower ibuprofen metabolic pathway, several of which were located on the same megaplasmid as the *ipf* genes, as well as others on the chromosome [7]. Another ibuprofen mineralizing strain belonging to *Sphingobium yanoikuyae* was successfully applied in bioaugmentation studies for the degradation of this contaminant in constructed wetlands planted with reeds [8]; however, the genetic background for its capacity to degrade ibuprofen remains unknown.

On the other hand, several bacterial isolates which do not belong to the bacterial family of *Sphingomonadaceae* were shown to fortuitously (and/or partially) oxidize (transform) ibuprofen to mono-, di- and/or trihydroxyibuprofen. One of these is the *Variovorax* sp. strain Ibu-1 which grows scarcely with this contaminant, generating a relatively low molar yield of biomass production, likely due to the only partial catabolism of this molecule [9]. Ibuprofen was also shown to be oxidized to mono- or oligohydroxy derivatives by bacterial isolates in the presence of alternative growth-supporting co-substrates (such as glucose, for example). These fortuitous oxidative catabolic reactions were probably based on cytochrome P450 systems or other types of generally unspecific oxidases. Some of the bacterial strains which co-metabolized ibuprofen when growing on complex media include the actinobacterial strain *Patulibacter medicamentivorans* [10], proteobacterial *Citrobacter* strains [11], and the *Firmicutes* strain *Bacillus thuringiensis* B1 [12]. The respective authors proposed, upon GC-MS or LC-MS analyses of their culture broths, a number of structures of potential products such as hydroquinone, hydroxyquinol, and (oxidated) dealkylated ibuprofen, and also products of the possible cleavage of the aromatic ring. However, a directed catabolic pathway for channeling these products into the central metabolism/Krebs cycle by specific catabolic enzymes was never confirmed. An early report indicated an initial reductive reaction of the carboxylic group of ibuprofen by a *Nocardia* species belonging to *Actinobacteria* [13]. The implementation of a species of *Micrococcus yunnanensis* into a model system was reported [14] from which the authors proposed a catabolic pathway based on structures deduced from TOF mass spectra obtained from a culture medium. A similar approach was chosen for the depletion of NSAIDs, including ibuprofen, upon the introduction of a *Pseudoxanthomonas* strain [15] into a similar system. 

Reports on the depletion of ibuprofen from real wastewater model systems or from laboratory model systems are numerous, and many of these studies focus on pharmaceutical and personal care products. Apart from real mineralization (complete degradation), the depletion of ibuprofen within such systems is often due to the fortuitous formation of hydroxylated derivatives [16], which may then undergo polymerization to bound residues and thus escape physico-chemical analyses [17,18]. 

Not only ibuprofen, a derivative of 2-phenylpropanoic acid, is of interest as a potent analgesic (see above), as 3-phenylpropanoic acid (3PPA, also known as 3-phenylpropionic or hydrocinnamic acid) and its derivatives have long been the subject of interest of the pharmaceutical, food, cosmetics, and technical industries [19]. Interestingly, 3PPA is the constituent of a large range of cosmetics and food additives because of its odor, and it acts as a technical emulsifier. Additionally, 3PPA has been found to act as an antimicrobial agent against numerous bacteria and fungi, and for this reason it is employed as a preservative [20,21]. On the other hand, its activity against bacteria and fungi may have some detrimental effects on the efficiency of biodegradation by the microbial flora in systems such as WWTPs and thus be of relevance for the safe production of drinking water. In fact, phenylpropanoic acid and structurally related compounds such as phenylacetic acid have been reported to be present in wastewater treatment systems and to possibly have negative impacts, especially on anaerobic processes [22,23,24].

In this work we aimed at obtaining a better understanding of the catabolic routes used by bacteria to degrade ibuprofen and 3PPA, the role of horizontal gene transfer and the relevance of these processes for contaminant elimination in WWTPs. To achieve this, we isolated ibuprofen degrading bacteria from a WWTP effluent and looked for the presence of *ipf* genes. We investigated the possible horizontal gene transfer of these genes from the newly isolated strains as well as from *S. granuli* RW412. How RW412 and the new strains catabolize 3PPA was also determined.

## 2. Materials and Methods

### 2.1. Chemicals

The chemicals used as substrates or standards were catechol, propanoic acid, benzoic acid, phenyllactic acid, phenylacetic acid, 2-phenylpropanoic acid, 3-phenylpropanoic acid (3PPA), *m*-tolylacetic acid, *p*-tolylacetic acid, ibuprofen as sodium salt, diclofenac, ibufenac, naproxen, and ketoprofen, and were all obtained from Sigma-Aldrich (Steinheim, Germany). Other chemicals used were 3-fluorocatechol (Alfa Aesar, Karlsruhe, Germany), and cinnamic acid (Fluka, Buchs, Switzerland). HPLC-grade solvents were supplied by Scharlau (Barcelona, Spain); all other chemicals were of analytical purity.

### 2.2. Enrichment, Isolation and Growth

Effluents from the same Granada WWTP described in a previous study [3] were enriched with 1 mM ibuprofen sodium salt (Sigma-Aldrich, Steinheim, Germany). Subcultivations of these enrichments were made in a Mineral Salts Medium (MSM) [25] with 1 mM ibuprofen at 30 °C on a rotary shaker at 100 rpm, and after a period of about two months several strains were isolated after streaking on solid MSM with Difco^®^ Agar Noble (BD, Le Pont de Claix, France) and 1 mM ibuprofen as the sole carbon and energy source and incubation at 30 °C. Colonies with different morphologies were transferred to new agar plates with 1 mM ibuprofen. Colonies growing on rich LB medium [26] for a purity check were re-transferred as isolated strains to MSM solid (for storage at 4 °C) and liquid media with 1 mM ibuprofen as the sole carbon and energy source and incubated at 30 °C on a rotary shaker at 200 rpm. Bacterial growth was monitored by optical density (OD_600nm_) on a Shimadzu UV-2401 PC spectrophotometer and the depletion of growth substrates was determined by HPLC (high performance liquid chromatography, see Section 2.7).

### 2.3. Characterization of the Isolates

To characterize each isolate, total genomic DNA was extracted from pure cultures of each strain using the Wizard Genomic DNA Purification Kit (Promega, Madison, WI, USA). To identify the isolates, the 16S rRNA gene was amplified by PCR using GM3F and GM4R primers (see Appendix A). Subsequently, the amplification product was subcloned into pGEM-T (Promega, Madison, WI, USA) and the gene product was sequenced using UNIV and REV primers (See Appendix A) at the Unidad de Genómica of the IPBLN (Granada, Spain). DNA sequences were analyzed by BLAST [27].

To determine the presence of *ipf* genes or other determinants of plasmidic regions in the isolates or transformants, a PCR was performed with KAPA2G Fast DNA Polymerase (KAPA Biosystems, Cape Town, South Africa) and the gene specific primers listed in Appendix A.

### 2.4. CoA Ligase Activity of the IpfF Protein with Ibuprofen-like Substrates

To detect the CoA activity of IpfF with different substrates, IpfF was purified in the same manner as described by Aguilar-Romero et al. [3]. The assays were performed with 1 mg purified IpfF in 50 mM Tris-HCl pH 7.5, with 0.5 mM substrate, 0.5 mM Coenzyme A (Sigma-Aldrich, Steinheim, Germany), 1 mM ATP (Sigma-Aldrich, Steinheim, Germany), 2 mM MgCl_2_, and 0.2 mM DTT. The enzymatic reactions were performed at room temperature (25–27 °C) for 35–58 min and the depletion of substrates and the formation of products were analyzed by HPLC (see Section 2.7).

### 2.5. Resting Cell Assays

Resting cell assays consisted of *S. granuli* RW412 or *R. wittichii* RW421 grown in MSM with either 2 mM ibuprofen or 3PPA as the sole carbon and energy source until an OD_600_ of approximately 1 was reached. Cells were centrifuged and washed twice with MSM and then resuspended in 2 mL of MSM with 2 mM 3PPA. An amount of 0.2 mM 3-fluorocatechol, a known inhibitor for the cleavage of catechols by respective dioxygenases, was added [28], and the conversion of 3PPA was followed by HPLC analysis; appearing peaks of accumulating catabolites were identified with the help of our in-house HPLC database (see Section 2.7).

### 2.6. Horizontal Gene Transfer

To determine whether the *ipf* genes are transmissible either as a genetic island or within the plasmid pRW412a [3], conjugation experiments were performed between the ibuprofen mineralizing strain *S. granuli* RW412 [3] and the dibenzofuran and dibenzo-*p*-dioxin, but not ibuprofen or 3PPA, mineralizing strain *Rhizorhabdus* (former *Sphingomonas*) *wittichii* RW1 [29] or the metabolically versatile but not ibuprofen utilizing *Pseudomonas putida* KT2440 [30], which is resistant to chloramphenicol (Cm). In the case of RW412 and *P. putida* KT2440, matings consisted of collecting biomass from each strain grown overnight on solid LB plates and mixing equal amounts onto a sterile filter on a fresh LB plate followed by incubation at 30 °C overnight. Biomass on the filter was resuspended in liquid MSM and serial dilutions were plated on MSM with Cm and 1 mM ibuprofen to select for KT2440 transconjugants. Alternatively, plasmidic DNA was obtained from RW412 by extraction of total DNA, as above, and then performing gel electrophoresis on agarose 0.8% and isolating DNA from the plasmidic DNA fraction using the QIAQUICK Gel Extraction Kit (Qiagen, Hilden, Germany). The resulting DNA was then introduced into electrocompetent cells of RW1 or KT2440 by electroporation [31], followed by serial dilutions on MSM and ibuprofen 1 mM.

On the other hand, for *R. wittichii* RW1, another conjugation approach was used by co-cultivating RW412 and RW1 in liquid MSM containing both carbon sources (1 mM ibuprofen and 1 mM dibenzofuran) over a period of about two months with subcultivations upon significant growth, followed by subcultivation on 1 mM ibuprofen for six more weeks. The culture medium was analyzed by phase contrast microscopic investigation, and isolates were obtained by plating on 1 mM ibuprofen-containing solid MSM, and on solid MSM containing 1 mM ibuprofen plus 20% (vol) of LB liquid medium.

Alternatively, this same conjugation approach was performed between RW422, RW423 and RW424 with *R. wittichii* RW1 in MSM with dibenzofuran and ibuprofen (1 mM each) and subculturing for more than 3 months.

### 2.7. Analytical Techniques

Assays of substrate consumption and transformation of (aromatic) organic compounds were determined by high-performance liquid chromatography (HPLC) on a Hewlett Packard (Waldbronn, Germany) 1050 HPLC-DAD system, with a Waters (Milford, MA, USA) Nova-Pak C18 HPLC column (5 µm, 3.9 × 150 mm) and a gradient of acidified water (1 mL of *ortho*-phosphoric acid per liter, eluent A) and correspondingly acidified (aqueous) acetonitrile (eluent B), with a flow rate of 0.85 mL/min. The gradient program was as follows: 2 min 100% eluent A followed by a gradient of 0–100% eluent B for 25 min. A total of 100% of eluent B was maintained for 10 min and then re-equilibrated in 100% eluent A for 9 min. Under these conditions, the following retention times were obtained (min in parenthesis): Catechol (3.61), 3-fluorocatechol (4.78), phenyllactic acid (7.39), phenylacetic acid (7.93), 3-phenylpropanoic acid (3PPA, 10.37), cinnamic acid (10.87), and ibuprofen (15.05).

Products obtained in vitro with different substrates and IpfF were identified using the same HPLC-DAD apparatus and column but with the eluents and gradient program as described previously [3]). The signals were analyzed with HPLC ChemStation software (Rev. A.02.05, Hewlett Packard). To determine substrate concentrations, peak areas at 210 nm were compared in the reaction before adding the IpfF protein and after the reaction time period had elapsed.

To determine the structural identification of the upcoming peak detected in cell cultures of RW412 growing at the expense of 3PPA, RW412 was grown in MSM with 3 mM 3PPA until the late exponential phase in an upscaled culture of 500 mL, followed by HPLC analysis. After the almost complete turnover of 3PPA, cells were removed by centrifugation and the supernatant was filtered through a 0.45 mm PTFE syringe filter. The supernatant was then acidified by the addition of HCl to pH 2 prior extraction with ethyl acetate. The organic phase was dried over anhydrous sodium sulfate and concentrated to dryness in a vacuum. An aliquot was re-dissolved in acetonitrile for a further HPLC analysis and the solid product from organic solvent extraction was subjected to nuclear magnetic resonance (NMR) spectroscopy. Spectra (^1^H, ^13^C, HSQC, and HMBC spectra) were acquired on a Bruker Avance Neo 600 MHz spectrometer at room temperature in deuterated chloroform. 

## 3. Results and Discussion

### 3.1. Characterization of Ibuprofen Mineralizing Consortia

In our previous study, we showed that bioaugmentation with the strain *S. granuli* RW412 of artificially contaminated secondary effluents from a Granada WWTP resulted in a significantly enhanced rate of elimination of ibuprofen; although at the end of a period of seven days of acclimation of the autochthonous flora in the WWTP effluent, the rate of ibuprofen elimination was relatively similar to that of the bioaugmented conditions [3]. Similarly, a qPCR of the *ipfF* gene, which encodes the first catabolic reaction of ibuprofen degradation which is the esterification of ibuprofen with coenzyme A, indicated the presence of this gene in the autochthonous bacterial population of the secondary effluent [3]. These observations motivated us to evaluate and re-confirm the worldwide ubiquity of ibuprofen-degrading microorganisms, which to date probably all belong to *Sphingomonadaceae*. 

Enrichment cultures inoculated with secondary effluent from a Granada WWTP were performed with ibuprofen as the sole carbon and energy source. Isolation from a subcultivation which actively grew with ibuprofen revealed two opaque isolates, RW422 and RW423, respectively, which showed swarming on the surface of the agar medium, whilst a third isolate, RW424, grew as pinpoint colonies on these agar plates. All of them were identified as belonging to *Pseudomonas citronellolis* upon analyses of their 16S rRNA genes by BLAST. Specifically, RW422 shares 99.8% sequence identity with the 16S rRNA genes of *P. citronellolis* strain C12; RW423 shares 100% identity with *P. citronellolis* strain G5.80; and RW424 shares 99.9% identity with *P. citronellolis* strain G5.80. None of the isolates were capable of growth in liquid MSM with 1 mM ibuprofen as the sole carbon and energy source, but the combinations of RW422 with RW424 (generation time (td) of 25 h) or of RW423 with RW424 (td of 25.2 h) showed good growth of the cell suspension, as well as the mixture of all three strains together. The biomass yield, based on the measurement of the OD_600_ on 1 mM ibuprofen, however, was relatively low, at 0.21 and 0.20, respectively, which could be explained by the production of numerous by-products upon analyses of the culture supernatants by HPLC. None of the peaks were identified when comparing their UV spectra with our in-house database, although several of them showed some significant similarity with catecholic compounds such as L-DOPA and *ortho*-hydroxy-substituted aromatic acids.

Up to now the only catabolic genes unequivocally shown to be involved in ibuprofen mineralization are the *ipf* genes, which are highly conserved in strains from three different worldwide locations [3,5,6]. PCR was used to detect for the presence of these genes in the isolates which resulted in the detection of *ipfABD* and *F* genes in both RW422 and RW423, while none could be detected in RW424 (Appendix A). Although *ipf* genes and similar genes have been identified in strains belonging to *Sphingomonadaceae*, *Xanthomonadaceae*, *Rhodospirilales* and *Comamonadaceae* [32], this is the first report in which these genes have been detected in *Pseudomonas* strains. Moreover, this is the first indication that these genes are required for ibuprofen mineralization outside of previously described *Sphingomonadaceae* strains. Nonetheless, the presence of *ipfABDEF* does not suffice, as neither RW422 nor RW423 can mineralize ibuprofen without the presence of RW424, which does not harbor these genes. This indicates that RW422 and RW423 probably require downstream enzymes to break down 4-isobutylcatechol and/or other ibuprofen catabolites, or that RW424 provides a required metabolite to RW422 or RW423 to permit the complete mineralization of ibuprofen. An experiment with filter-sterilized spent culture medium from RW424 grown in MSM with propionate did not permit RW422 or RW423 to grow with ibuprofen as the sole carbon or energy source. Therefore, the required metabolite or other capacity provided by RW424 to the consortia to permit for ibuprofen mineralization remains unclear. However, as *ipfABDEF* genes have been associated with megaplasmids containing additional *ipf* genes involved in ibuprofen metabolism [3,6,7], and the importance of genetic mobile elements needs to be determined for these strains. 

### 3.2. Horizontal Transfer of ipf Genes

Operons of genes for the catabolism of organic environmental pollutants or for the detoxification of toxic heavy metals are often organized as genomic islands [33], some of which are almost generally located on megaplasmids [34]. As described previously, the *ipf* genes found in ibuprofen mineralizing strains are bordered by IS elements [3,5,6]. Moreover, they have been associated to megaplasmids in two different isolates [3,6], although always in bacteria belonging to the *Sphingomonadaceae* family. This megaplasmid appears to be autotransmissible within *Sphingomonadaceae* species, as shown by Aulestia et al. [6] by the transfer of pIBU218 from *R. wittichii* strain MPO218 to the related bacterium, *S. granuli* TFA. This motivated us to evaluate whether the *ipf* genes of *S. granuli* RW412 [3] are also transferrable to *Sphingomonadaceae* strains or to other bacteria. As a possible non *Sphingomonadaceae* recipient strain, the metabolically versatile *P. putida* KT2440 [30] was considered. However, neither conjugation nor electroporation with the plasmidic DNA fraction of *S. granuli* RW412 resulted in *P. putida* KT2440 derivatives capable of growth with ibuprofen. As a *Sphingomonadaceae* recipient, the dibenzofuran and dibenzo-*p*-dioxin, but not ibuprofen, mineralizing strain *R. wittichii* RW1 [29] was considered. In this case, electroporation with the plasmidic DNA fraction of RW412 into electrocompetent RW1 cells did not result in ibuprofen utilizing RW1 derivatives. However, conjugation between strain RW412 and RW1 as a co-culture followed by a long period of enrichment and selection resulted in an isolate which was morphologically similar to RW1 by microscopic analysis and conserved growth on dibenzofuran and dibenzo-*p*-dioxin characteristic of this strain, but which additionally had acquired the capacity to grow with ibuprofen as the sole carbon and energy source. The obtained isolate, termed RW421, showed a generation time (td) of 5.4 h during growth with ibuprofen, which is significantly faster growth than the “genetic” parent RW412, which exhibited a generation time of 16 h [3]. The biomass yield with 1 mM ibuprofen was 0.472 of OD_600_. The sequencing of the 16S rRNA gene of RW421 confirmed its identity as *R. wittichii* RW1 (99.93% identity). The presence of the *ipf* operon was confirmed by PCR (Appendix A) and, therefore, it is likely that, analogous to the study by Aulestia et al. [6], the RW421 received the pRW412a megaplasmid. To investigate this, PCR was performed with representative genes for different plasmid regions of pRW412a. The results (Appendix A) show that RW421 has at least part of three of the four regions associated with *ipf* genes specifically, region III with the *ipfABDEF* genes, region II with *ipfI*, and region IV with *ipfH* and aromatic degradation genes, but not region I with *ipfM* [7] or region V+I with plasmid conjugation and replication genes. Therefore, it appears that either only a portion of plasmid pRW412a was transferred from *S. granuli* RW412 to *R. wittichii* RW1, or that parts were lost during co-culture and enrichment, a phenomenon observed previously for the similar plasmid pIBU218 [6,7]. However, since the sequencing of *R. wittichii* RW421 was not performed, exactly which parts of pRW412a remain in RW421 is unknown. Nonetheless, since RW421 can mineralize ibuprofen, the absence of some of the lower pathway activities encoded on the original plasmid such as the hydroxymuconate semialdehyde dehydrogenase activity encoded by *ipfM* may have been replaced by activities encoded in the recipient RW1 genome such as by a gene which shares more than 62% identity with *ipfM*. From these experiments it appears that the stability of pRW412a in *Pseudomonas* or *Rhizorhabdus* is initially poor if ibuprofen mineralization is used for selection, perhaps due to the additional requirements for ibuprofen mineralization from the receptor strain, which requires not only the presence of all the genetic capabilities but also the sufficient adaptation for proper gene expression and coordination. 

The fact that *ipf* genes were detected in the *P. citronellolis* species RW422 and RW423 suggests that the horizontal transfer of potential *ipf* genes is not limited to occur only between some *Sphingomonadaceae* species. Detection by PCR (Appendix A) of elements related to the *ipf* encoding megaplasmids from *S. granuli* RW412 or *R. wittichii* MPO218 revealed that both *P. citronellolis* RW422 and RW423 contain all of the elements tested for, suggesting that they contain the entire or, at least, a large part of the ibuprofen-related catabolic plasmids previously only associated with the *Sphingomonadaceae* species. We intended to determine whether this *ipf* containing mobile element could be transferred to the same *R. wittichii* RW1 used to obtain *R. wittichii* RW421, again relying on the capacity of the latter to mineralize dibenzo-*p*-dioxin and dibenzofuran, but not ibuprofen. For this purpose, *R. wittichii* RW1 was grown in combination with RW422, RW423, or RW424 in MSM with dibenzofuran plus ibuprofen. The results from these co-culturing trials showed that RW422 as well as RW423 yielded significant growth in co-culture together with RW1, but the combination of RW424 with RW1 did not. However, upon the subculturing of RW422 and RW1, or RW423 and RW1 for more than three months, we were not able to isolate a single derivative of RW1 which could grow with ibuprofen as the sole carbon and energy source. This suggests that either no stable transfer occurred or that if it did the group of *ipf* genes transferred were insufficient for full mineralization. On the other hand, the fact that RW1 can replace the function of *P. citronellolis* RW424 in consortia with RW422 or RW423 suggests that RW424 and RW1 share the same or similar characteristics missing in RW422 or RW423 to permit the full mineralization of ibuprofen.

### 3.3. Catabolism of 3-Phenylpropanoic Acid (3PPA)

The catabolism of 3PPA has only been elucidated for the *E. coli* strain K12 [35,36]. In this pathway, the degradation of this compound was initiated through attack by a dioxygenase at the positions 2 and 3, forming a dihydrodiol which was subsequently rearomatized to 2,3-dihydroxyphenylpropionate, followed by dioxygenolytic *meta*-cleavage before the formed catabolites were channeled into the TCA/Krebs cycle for complete mineralization. The genes responsible for these catabolic reactions by *E. coli* were identified and characterized a couple of years later [36]. However, the reported catabolism of phenylacetic acid [37] and ibuprofen initiates by the esterification of the carboxylic group with coenzyme A (CoA).

*Sphingomonadaceae* species exhibit unique catabolic features for different classes of environmental contaminants such as tetralin [38], PAHs [39], HCHs [40], and diaryl ethers such as dibenzo-*p*-dioxin [29], as well as ibuprofen [3,4,6]. Therefore, we sought to re-evaluate the catabolism of 3PPA in *Sphingomonadaceae* strains. We had already reported that *S. granuli* RW412, apart from ibuprofen, uses 3-phenylpropanoic acid (3PPA, also known as 3-phenylpropionic or hydrocinnamic acid) and several structurally related aromatic acids, such as ibufenac (2-[4-(2-methylpropyl)phenyl]acetic acid), phenylacetic acid, 4-methylphenylacetic acid (*p*-tolylacetic acid) and 2-(4-methylphenyl)propanoic acid (α,4-dimethylphenylacetic acid), as the sole carbon and energy sources for growth [3]. As the catabolism of ibuprofen and phenylacetic acid [37] initiates by the esterification of the carboxylic group with CoA, we were interested in determining if such activation by CoA would also apply for 3PPA, as well as for this class of compounds such as phenylacetates and -propanoids, which are widely present in nature. For this, we performed in vitro assays with the purified CoA ligase of the ibuprofen mineralizing strain RW412, IpfF, with 3PPA as a substrate. The formation of a new peak was observed by HPLC, while that of 3PPA decreased (Appendix A). Analogous to that observed for the ibuprofenyl-CoA product, the UV spectrum of this new peak was similar to that of CoA, indicating it to be 3-phenylpropanoyl-CoA (3PPA-CoA) (Appendix A). The CoA ligase activity of IpfF was also tested with other phenylacetates and -propanoids and compared to its activity with ibuprofen. The results showed that compared to the 88% conversion of ibuprofen to ibuprofenyl-CoA, 57% of 3PPA was converted to 3PPA-CoA (Table 1). With respect to NSAIDs such as ibufenac, diclofenac, naproxen and ketoprofen, only ibufenac served as a good substrate for IpfF to form its corresponding CoA derivative. On the other hand, phenylacetic acid and methyl or dimethyl derivatives also were good substrates while 2-phenylpropanoic acid and 3PPA were converted to their corresponding CoA derivatives by IpfF to a lesser extent.

Next, we sought to determine whether the expected intermediates from 3PPA-CoA catabolism by *ipf* gene products or other metabolites would appear as RW412 grows with 3PPA. When growing RW412 on 3PPA, HPLC gradient analysis of the culture revealed that amongst small amounts of several polar metabolites, a large amount of a less polar metabolite eluting later than 3PPA accumulated. This major compound was not 3PPA-CoA as its UV spectra did not share any resemblance to that of the IpfF product. Moreover, this compound accumulated but then disappeared, as 3PPA was consumed and RW412 continued to grow (Figure 1).

In order to identify the accumulating compound of RW412 growing with 3PPA, it was extracted from the culture medium and analyzed by ^1^H and ^13^C NMR (Figure 1 and Appendix A). As shown in Figure 2, the spectrum and comparison with an NMR spectrum of commercial/authentic 3PPA (https://orgspectroscopyint.blogspot.com/2017/07/3-phenylpropionic-acid.html (accessed on 29 April 2022), and https://pubchem.ncbi.nlm.nih.gov/compound/3-Phenylpropionic-acid#section=1H-13C-NMR-Spectra (accessed on 29 April 2022), unambiguously showed the compound to be 3-phenyl-2,3-propenoic acid (cinnamic acid). Specifically, the spectrum in Figure 2 showed that the coupling of 16 Hz of the additional signals at 6.47 and 7.81 ppm (signals between 2.6 and 2.9 ppm indicative for the methylene protons were not present any more), confirm the presence of the double bond with *trans* configuration of the isolated metabolite and, therefore, determining a mass spectrum of the isolated metabolite was no longer necessary for structure identification.

In order to further discern the pathway used by *S. granuli* RW412 for 3PPA mineralization, other more polar peaks detected in the growth medium of this strain with 3PPA were determined. The degradation of 3PPA by *E. coli* was originally shown to proceed via the catabolite 2,3-dihydroxyphenylpropionate [35,36], a compound which is not commercially available. Therefore, we used *E. coli* MC1061 [36] for its experimentally confirmed production by growing this strain on 3PPA in MSM and identifying the accumulated polar product as 2,3-dihydroxyphenylpropionate, which elutes significantly earlier than 3PPA when the medium was analyzed by HPLC. We found that the retention time and UV spectrum of one of the small polar peaks of RW412 growing with 3PPA was identical to that of the 2,3-dihydroxyphenylpropionate produced by *E. coli* MC1061, indicating that RW412 uses, at least partially, this pathway for 3PPA biotransformation. Although the genome of RW412 does not harbor genes with high identity with the *hca* genes of *E. coli* MC1061 responsible for this pathway [36], perhaps some of the 6 dioxygenases encoded by RW412 [3] may fortuitously catabolize this type of biotransformation.

Another small polar peak which appeared as the cinnamic acid peak depleted in the culture medium of RW412 growing with 3PPA could be identified as 3-phenyllactic acid, as the UV spectrum and retention time correspond to the pure standard compound. The identification of this peak and of cinnamic acid leads us to propose a pathway which includes the reverse oxidation reaction of the currently only known enzyme to be involved in 3PPA metabolism by *Clostridia*, cinnamate reductase [41], and therefore would present a novelty (Figure 3). On the other hand, as we have shown that IpfF catalyzes the esterification of 3PPA with coenzyme A to yield its CoA ester, it cannot be eliminated as a possibility that this product undergoes dehydrogenation (reduction) of the side chain to yield cinnamoyl-CoA, and which could be further channeled into the gentisic acid pathway as proposed for the archeon *Haloferax* [42] or converted to cinnamic acid. However, since CoA derivatives were not observed, and such a series of reactions would be much costlier metabolically for efficient growth, it is more likely that the direct reduction of 3PPA to cinnamic acid followed by hydration to phenyllactic acid occurs. Nonetheless, in order to determine whether the 3PPA-CoA product would be further converted to catechol by the *ipf* genes, resting cell assays were performed with 3PPA using RW412 cells grown previously either in ibuprofen or in 3PPA and then inhibited for further ring cleavage by the addition of 3-fluorocatechol [28]. The products detected by HPLC from 3PPA with ibuprofen grown cells were cinnamic acid, catechol and a PPA-like compound. However, when 3PPA grown cells were used, the products detected from 3PPA were cinnamic acid, phenyllactic acid, and a PPA-like compound, but not catechol. These results indicate that the *ipf* operon is likely to be downregulated in 3PPA grown cells, but when upregulated in ibuprofen grown cells are capable of converting the 3PPA to catechol previous to ring cleavage and metabolism. Irrespective of *ipf* induction, 3PPA is converted to cinnamic acid, indicating this pathway to be the main route used by *S. granuli* RW412 to degrade 3PPA.

In order to determine 3PPA degradation by the other strains described in this work, the growth with this compound and the possible metabolites produced were analyzed for *R. wittichii* RW1, its ibuprofen mineralizing derivative *R. wittichii* RW421, and the isolates *P. citronellolis* RW422, RW423 and RW424 (either individually or as two strain consortia). Whilst RW1 cannot use 3PPA for growth, its derivative RW421 could mineralize 3PPA. According to HPLC analyses, RW421 also accumulates transiently cinnamic acid as well as a more polar compound with a UV spectrum and retention time identical to the 2,3-dihydroxyphenylpropionate produced by *E. coli* MC1061. Resting cell assays performed with 3PPA using RW421 cells grown previously either in ibuprofen or in 3PPA and then inhibited with 3-fluorocatechol produced cinnamic acid in both cases, and sometimes a 3PPA-like compound, but catechol accumulation was not observed. This suggests that in RW421, the *ipf* operon is more quickly silenced than in RW412, or that 3PPA-CoA production or conversion to catechol requires additional elements no longer present in the pRW412a megaplasmid it received from RW412. 

On the other hand, neither *P. citronellolis* RW422, RW423 nor RW424 could grow with 3PPA alone. However, two-strain consortia RW422/RW424 and RW423/RW424 could mineralize 3PPA. These consortia excrete cinnamic acid and still unknown catechol-like intermediates such as L-DOPA-like compounds. The consortium of RW1 and RW422 can also mineralize 3PPA and was observed to accumulate minute amounts of cinnamic and phenyllactic acid. Altogether these results suggest that part of the capacity for 3PPA mineralization is encoded by genes on the pRW412a-like megaplasmid, while other elements are encoded in the chromosome of *R. wittichii* RW1 or *P. citronellolis* RW424. Nevertheless, all 3PPA mineralizing bacteria or consortia use the cinnamic acid pathway. From the ecotoxicological point of view, cinnamic acid and its derivative, phenyllactic acid, have only been described to have some antimicrobial effects [43,44]. Moreover, these derivatives are produced transiently by 3PPA mineralizing bacteria or consortia. Therefore, increasing bacterial populations which use this pathway to mineralize 3PPA is a useful and safe strategy which could be studied in more detail to eliminate this phenylpropanoid from WWTPs.

## 4. Conclusions

The results obtained in this study increase the knowledge of the degradation of the PPCP phenylpropanoids ibuprofen and 3PPA by bacteria from WWTPs. New biodegrading bacterial strains different from *Sphingomonadaceae* species which, as a consortium, can mineralize ibuprofen have been identified. We show that this capacity is associated with the presence of the upper ibuprofen degradation pathway *ipf* genes linked to megaplasmids and present evidence for the horizontal gene transfer of these genes between *Sphingomonadaceae* strains. In addition, we describe a new catabolic pathway for 3PPA mineralization involving cinnamic acid as a major intermediate that could contribute to improve its elimination. However, the enzymes directly involved in this pathway and gene regulation require further investigation. Future prospects should focus on strategies to increase ibuprofen and 3PPA mineralizing bacterial populations in WWTPs as well as studies of the interactions between members of these populations for the efficient elimination of ibuprofen and 3PPA from water resources.

## Figures and Tables

**Figure 1 genes-14-00442-f001:**
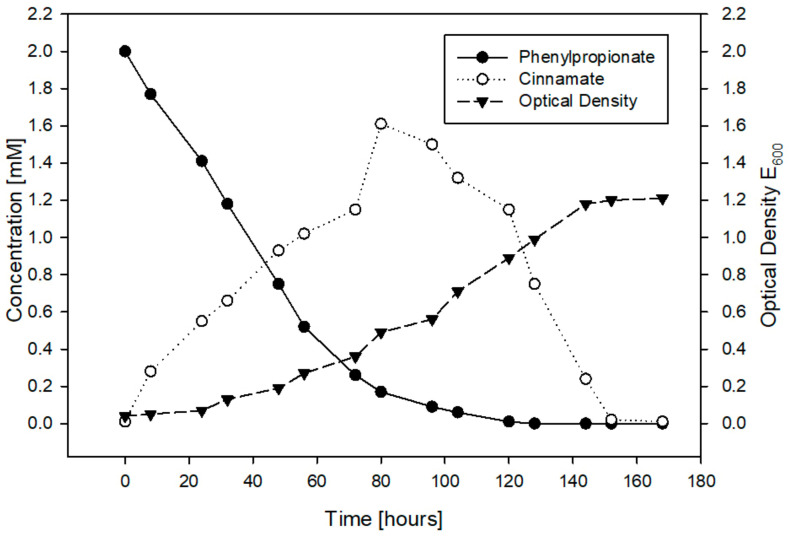
Growth of *Sphingopyxis granuli* RW412 as indicated by the depletion of 3PPA (solid circles) over time and the increase in biomass (solid triangles), as determined by measuring the Optical Density (E_600_) of the culture. The accumulation and depletion of (later-on identified) cinnamate is indicated with open circles.

**Figure 2 genes-14-00442-f002:**
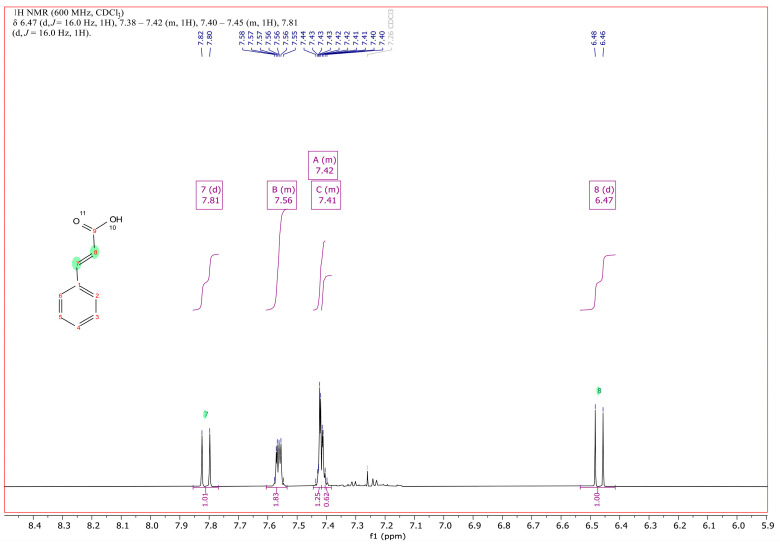
^1^H NMR spectrum of the double bond of the prop-2-enoic moiety of cinnamic acid at 6.47 and 7.81 ppm and of its aromatic protons from 7.41 to 7.56 ppm, recorded at 600 MHz in deuterochloroform. The minute signals around 7.3 ppm correspond to the aromatic protons of the biologically converted parent 3PPA.

**Figure 3 genes-14-00442-f003:**
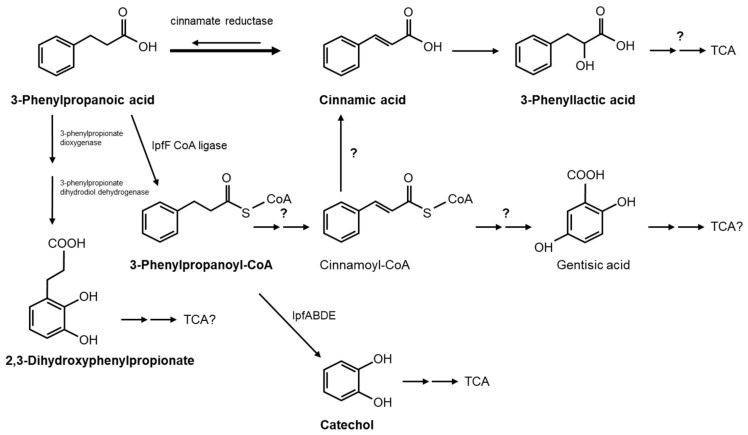
Possible degradation pathways of 3PPA by *S. granuli* RW412. The compounds identified in this study, which were 3PPA, cinnamic acid, phenyllactic acid, 3-phenylpropanoyl-CoA, 2,3-dihydroxyphenylpropionate, and catechol are indicated in bold.

**Table 1 genes-14-00442-t001:** Conversion of 0.5 mM substrate by IpfF, a CoA ligase.

Substrate	Product	Estimated Concentration CoA Product (mM) ^1^
Ibuprofen	ibuprofenyl-CoA	0.44
Diclofenac	diclofenacyl-CoA	0.04
Ibufenac	ibufenacyl-CoA	0.41
Naproxen	naproxenyl-CoA	0.12
Ketoprofen	ketoprofenyl-CoA	0.00
Benzoic acid	benzoyl-CoA	0.00
Phenylacetic acid	kphenylacetyl-CoA	0.35
2-Phenylpropanoic acid	2-phenylpropanoyl-CoA	0.27 *
3-Phenylpropanoic acid (3PPA)	3-phenylpropanoyl-CoA	0.28 *
*m*-Tolylacetic acid	*m*-tolylacetyl-CoA	0.42 *
*p*-Tolylacetic acid	*p*-tolylacetyl-CoA	0.35 *
α,4-Dimethylphenylacetic acid	α,4-dimethylphenylacetyl-CoA	0.42

^1^ Estimated by amount of substrate removed from the reaction (HPLC). * Estimated from the reduction of total peak areas of CoA due to coinciding retention times with substrate.

## Data Availability

Not applicable.

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
