# Peer review of "Biodegradation of Microtoxic Phenylpropanoids (Phenylpropanoic Acid and Ibuprofen) by Bacteria and the Relevance for Their Removal from Wastewater Treatment Plants"

_genes, 2023, doi:10.3390/genes14020442_

Round 1

Reviewer 1 Report

-One thing that is not clearly addressed is in the discussion on the different mechanisms of biodegradation of microtoxic phenylpropanoids, are any of the endproducts a potential hazard to the environment and to living organisms?  If this is the case, are there any plans to remedy such a problem.

-Your discussion included how different strains of Sphingomonadaceae can laterally transfer the ipf genes responsible for the catabolism of these phenylpropanoids.  There seems to be implications that using a combination of these isolates may improve the efficiency of removing these contaminants.  Are there any plans to study such interactions and if so, how would one control the quality of such a consortium to obtain consistent results?

Author Response

We would like to thank the reviewer for the helpful and constructive comments which without a doubt have helped us to improve the manuscript. Please find below a point by point response to all the comments made by the reviewer. Reponses are indicated by “>” followed by text in bold. Page and line numbers refer to the final revised version of the manuscript.

-One thing that is not clearly addressed is in the discussion on the different mechanisms of biodegradation of microtoxic phenylpropanoids, are any of the endproducts a potential hazard to the environment and to living organisms? If this is the case, are there any plans to remedy such a problem.

>Mineralization pathways permit complete degradation of compounds to CO2 and for bacterial growth (biomass production). In the different mechanisms of biodegradation described, some of the possible intermediates, such as catechols, cinnamic acid and phenyllactic acid can have some toxicity or antimicrobial activity but only if they are not further degraded. Since, the major route used by 3PPA mineralizing bacteria appears to be via cinnamic acid and phenyllactic acid, their microtoxic activity is mentioned with references in the text (lines 500-502). However, since they are produced transiently by 3PPA mineralizing bacteria, promoting these bacteria could be a useful and safe strategy to eliminate this phenylpropanoid. This concept has been introduced into the text at lines 502-506.

-Your discussion included how different strains of Sphingomonadaceae can laterally transfer the ipf genes responsible for the catabolism of these phenylpropanoids. There seems to be implications that using a combination of these isolates may improve the efficiency of removing these contaminants. Are there any plans to study such interactions and if so, how would one control the quality of such a consortium to obtain consistent results?

>We have modified the conclusions section and included future plans regarding the study of these interactions (See lines 518-521).

Reviewer 2 Report

the manuscript, titled"Biodegradation of microtoxic phenylpropanoids (phenylpropanoic acid and ibuprofen) by bacteria and the relevance for their removal from wastewater treatment plants" is a very well-written piece of work. there are some general comments for the paper that if incorporated and addressed can make the manuscript even better. Please pay a little attention to the use of gene names throughout the manuscript and modify the conclusion and some parts in the introduction. you can find my comments in the PDF attached. 

Author Response

We would like to thank the reviewer for the helpful and constructive comments which without a doubt have helped us to improve the manuscript. Please find below a point by point response to all the comments made by the reviewer. Reponses are indicated by “>” followed by text in bold. Page and line numbers refer to the final revised version of the manuscript.

- the manuscript, titled"Biodegradation of microtoxic phenylpropanoids (phenylpropanoic acid and ibuprofen) by bacteria and the relevance for their removal from wastewater treatment plants" is a very well-written piece of work. there are some general comments for the paper that if incorporated and addressed can make the manuscript even better. Please pay a little attention to the use of gene names throughout the manuscript and modify the conclusion and some parts in the introduction. you can find my comments in the PDF attached.

Comments in the PDF:

- Towards the end of the introduction adding probably 5-6. lines giving a relevence of why this study is designed and what will be achieved and hoow, briefly. like giving a rationale will be of high importance and will make the ontroduction strong.

> Thank you for this suggestion. We added some lines to give the rational of the study (See lines 113-119).

- 2.1 last para graph from line 130-136 should be a sub section with names as materials may be or any other suitable one as it is merged with the classic experiment of isolation and enrichment. it should come beofr this section. may be mke it first section.

> This change was made in the Materials and Methods by adding a new subsection ‘2.1 Chemicals’ (Lines 122-129)

-Lines 150 onwards: make it small letters, ipf

-In IpfF remove extra F, please double check in the whole manuscript

> It appears that ipf genes in general and the ipfF gene product, ipfF, causes confusion. To make clear that IpfF is a protein which has CoA ligase activity, we have introduced changes on lines 158-160 and on line 220.

Conclusions section:

- Do you claim here that before your study it was thought that only sphingomonads are involved in the degradation of ibuprofen…

> Throughout the text we have mentioned that in literature only strains belonging to the Sphingomonadacaea family have been shown, up to now, to completely mineralize ibuprofen (see lines 55-57, 245-246, 273-275). In the text we previously used ‘sphingomonads’ to refer not only to Sphingomonas but to all Spingomonadaceae bacteria but perhaps this was not clear. In order to avoid any misunderstandings, we have replaced all references to ‘sphingomonads’ to ‘Sphingomonadaceae species’ or ‘-strains’ throughout the text as well as on lines 510-512.

- Kindly try to improve the conclusion and describe solid salient highlights of the study and emphsize your stregths and describe the way ahead.. additions to the pool of knowl;edge and clearly mention furture prospects

>We have improved the conclusions section by highlighting the major findings, their implications for the field and future prospects (See lines 509-521).